# Diagnosis and Prognosis of Canine Melanocytic Neoplasms

**DOI:** 10.3390/vetsci9040175

**Published:** 2022-04-06

**Authors:** Rebecca C. Smedley, Kimberley Sebastian, Matti Kiupel

**Affiliations:** 1Veterinary Diagnostic Laboratory, Michigan State University, Lansing, MI 48910, USA; sebast52@msu.edu (K.S.); kiupel@msu.edu (M.K.); 2Department of Pathobiology and Diagnostic Investigation, College of Veterinary Medicine, Michigan State University, East Lansing, MI 48910, USA

**Keywords:** melanoma, canine, dog, melanocytic neoplasm, Ki67, sentinel lymph node, prognosis, metastasis

## Abstract

Canine melanocytic neoplasms have a highly variable biological behavior ranging from benign cutaneous melanocytomas to malignant oral melanomas that readily metastasize to lymph nodes and internal organs. This review focuses on the diagnosis and prognosis of canine melanocytic neoplasms. While pigmented melanocytic neoplasms can be diagnosed with fine-needle aspirates, an accurate prognosis requires surgical biopsy. However, differentiating amelanotic spindloid melanomas from soft tissue sarcomas is challenging and often requires immunohistochemical labeling with a diagnostic cocktail that contains antibodies against Melan-A, PNL-2, TRP-1, and TRP-2 as the current gold standard. For questionable cases, RNA expression analysis for TYR, CD34, and CALD can further differentiate these two entities. The diagnosis of amelanotic melanomas will be aided by submitting overlying and/or lateral flanking epithelium to identify junctional activity. Wide excision of lateral flanking epithelium is essential, as lentiginous spread is common for malignant mucosal melanomas. Combining histologic features (nuclear atypia, mitotic count, degree of pigmentation, level of infiltration, vascular invasion; tumor thickness and ulceration) with the Ki67 index provides the most detailed prognostic assessment. Sentinel lymph nodes should be evaluated in cases of suspected malignant melanomas using serial sectioning of the node combined with immunohistochemical labeling for Melan-A and PNL-2.

## 1. Introduction

Melanocytic tumors are common in dogs and occur primarily in the skin and oral cavity, with oral melanocytic neoplasms comprising 30–40% of all canine oral neoplasms [1,2]. Although the diagnosis of pigmented melanocytic neoplasms is straightforward, amelanotic forms can pose a diagnostic challenge [3]. Furthermore, while cutaneous melanocytic neoplasms tend to be less aggressive than their oral counterpart and are often cured by surgical excision, both highly malignant cutaneous and low malignant oral melanocytic tumors occur with some frequency and accurate prognostication is essential for successful therapeutic intervention [4,5,6,7,8,9]. This review will provide an overview of the current state of the diagnosis and prognosis of canine melanocytic neoplasms. It is essential to define a number of terms that are commonly used for the diagnosis and prognosis of these neoplasms to ensure consistency in communication.

In dogs, the term melanocytoma is used to describe benign cutaneous melanocytic neoplasms, while the term malignant melanoma is used to describe malignant melanocytic neoplasms regardless of their location [7,10]. For routine surgical pathology, the term “melanocytic neoplasm” should be used when the histologic parameters alone cannot differentiate between a melanocytoma and a malignant melanoma [7,9]. The term melanosarcoma should be avoided, as it incorrectly classifies malignant melanomas as being derived from mesenchymal cells rather than the neural crest. In addition, the term nevus should not be used in dogs as it historically describes a birthmark in humans and has been used to identify both melanocytic and non-melanocytic non-neoplastic cutaneous proliferations in dogs [7,9,10].

There are also a number of important morphologic features that are characteristic of the different entities of melanocytic neoplasms in dogs. “Dermal” is the term used to describe melanocytic tumors that are confined to the dermis and do not involve the epidermis (Figure 1) [9,11]. In contrast, “compound” melanomas have both an epithelial and a subepithelial component. The subepithelial (dermal/submucosal) component causes the primary mass effect (Figure 2) [7,9,12]. This mass effect is referred to as vertical growth [7,12]. Compound melanomas are characterized by nests or individual neoplastic melanocytes in the basal layer of the epithelium, an important diagnostic criterion referred to as junctional activity (Figure 3) [7,11,12]. Such intraepithelial neoplastic melanocytes tend to migrate laterally among basal keratinocytes, which is referred to as lentiginous/lateral spread or radial growth [7,11,12]. If neoplastic cells are present within the more superficial layers of the epithelium, such growth is called pagetoid spread [7,11,12].

## 2. Diagnosis of Canine Melanocytic Neoplasms

Most melanocytic neoplasms are easily diagnosed via cytology or histopathology. Cytology has been shown to have a sensitivity and specificity of 100% for the diagnosis of pigmented melanocytic neoplasms; however, both sensitivity (67–100%) and specificity (85–100%) decrease when attempting to diagnose amelanotic melanocytic neoplasms [13,14,15]. Cytologically, well-differentiated melanocytes exfoliate in sheets, small clusters, or individually. They appear round, polygonal, or spindloid and have small amounts of pale gray to blue cytoplasm that contains melanin pigment that appears rod-shaped or finely granular and stains blue-green to golden to black with standard Romanowsky-type stains (Figure 4) [13,14]. Cytologic atypia is generally mild; nuclei are round to oval, finely stippled, and generally have one large, prominent nucleolus. In addition, melanin-laden macrophages (melanophages) may be variably present.

Poorly differentiated neoplastic melanocytes often have increased cytologic atypia such as increased anisocytosis and anisokaryosis, multinucleated cells, multiple and variably sized and shaped nucleoli, variable chromatin patterns, and variable nuclear-to-cytoplasmic ratios with cytoplasm that may contain few to many, small to large, clear vacuoles [13,14,15]. Neoplastic cells may lack, or have very little, melanin pigment, and it may be difficult to ascertain cell origin; therefore, a melanocytic neoplasm should be considered for any tumor that is difficult to classify cytologically.

Differential diagnoses include pigmented basal cell tumors, Schwannomas, and peripheral nerve sheath tumors. Immunocytochemistry (ICC) for Melan-A may be helpful to differentiate between these non-melanocytic tumors or to diagnose amelanotic melanocytic neoplasms. If Melan-A ICC is negative and amelanotic melanoma is still strongly suspected, histopathology is recommended. Histopathology generally allows for more accurate diagnosis, via assessment of other morphologic features and the ability to perform additional immunohistochemical markers, and also provides an accurate prognosis [15]. Alternatively, for some histologically amelanotic melanocytic neoplasms that do not exhibit other confirmatory features of melanocytic origin, cytology may be the more sensitive method for detecting melanin pigment. However, in some cases, melanin may not be seen cytologically either.

Histologically, well-differentiated neoplastic melanocytes present as round to polygonal to spindloid cells with variable amounts of eosinophilic cytoplasm that contains varying amounts of brown melanin granules. Different cellular morphologies have been reported, the most common being epithelioid/polygonal, spindloid/fibromatous, and mixed epithelioid and spindloid (Figures 5–7) [7,9]. Multiple cell morphologies and arrangements of neoplastic cells, such as packets, sheets, and bundles, are often present in the same neoplasm [7,9]. The nuclei of neoplastic cells are round to oval, finely stippled to vesiculate, exhibit minimal to mild anisokaryosis, and generally contain one single prominent centrally oriented nucleolus, creating an “owl’s eye” appearance [9,11]. An important diagnostic feature is the presence of neoplastic cells at the subepithelial stroma-epithelial junction. While junctional activity and pagetoid growth are strong diagnostic criteria, the presence of neoplastic cells just beneath the basement membrane, even without junctional activity, raises concern for a melanocytic neoplasm [7,9,11]. Identifying individual intraepithelial neoplastic melanocytes that are non-pigmented can be challenging. Individual intraepithelial melanocytes can either represent a neoplastic or hyperplastic lesion. Nuclear pleomorphism, anisocytosis, anisokaryosis, and retraction artifacts that leave a clear space around such intraepithelial cells are all features suggestive of neoplastic melanocytes [7].

While epithelioid/polygonal, spindloid/fibromatous, and mixed epithelioid and spindloid are still the most common cellular morphologies of poorly differentiated melanocytic neoplasms, other cellular morphologies can also be seen in these less differentiated tumors [7,9]. Such less common cell types and patterns include round cell, small cell, balloon cell (also known as a clear or signet-ring cell), whorled/dendritic, adenomatous/papillary, giant cell, angiotropic and angiomatoid/pseudovascular, and those with osteocartilaginous differentiation [7,9,16,17]. Poorly differentiated neoplastic cells often contain little to no melanin pigment. The nuclei often contain multiple distinct nucleoli, exhibit moderate to marked anisokaryosis, and have variable chromatin patterns, similar to what is seen cytologically [7,9,16].

Amelanotic melanocytic neoplasms are the most challenging to diagnose (Figure 8) [3,18]. Histologically, junctional activity may be the strongest diagnostic criterion, making it essential to submit intact epithelium overlying the neoplasm. If a lesion is severely ulcerated, submission of intact epithelium along the edges of the ulcer is still recommended, as lentiginous spread may be observed in the adjacent intact epithelium. If a diagnosis cannot be reached on routine microscopic examination, immunohistochemistry (IHC) for the melanocyte-specific markers Melan-A, PNL-2, tyrosine reactive protein (TRP)-1 and TRP-2 is most commonly used to confirm melanocytic differentiation [Figure 9] [3,18,19,20,21,22]. A melanocytic diagnostic cocktail that contains antibodies against Melan-A, PNL-2, TRP-1, and TRP-2 has been shown to improve the overall sensitivity to 93.9% when differentiating from soft tissue sarcomas, while maintaining 100% specificity, and represents the current gold standard for diagnosing amelanotic melanocytic neoplasms in dogs [3,18]. While the subepithelial neoplastic cells may be negative, intraepithelial nests of neoplastic cells are the most commonly positive cells detected by IHC, further emphasizing the importance of submitting specimens with intact epithelium [Figure 10] [12,18]. Thus, incisional and excisional biopsies should include overlying and lateral flanking intact epithelium to increase the likelihood of an accurate diagnosis. For the diagnosis of poorly differentiated neoplasms that lack junctional activity, it is often helpful to include antibodies against other targets to exclude other differentials. For example, if a carcinoma is a differential diagnosis, performing IHC for pancytokeratin (an epithelial cell marker) aids in diagnosis confirmation. Differentiating between canine oral amelanotic spindloid malignant melanomas and oral soft tissue spindle cell sarcomas is often the most challenging. Regardless, in a recent study, the melanocytic diagnostic cocktail accurately identified all amelanotic oral spindle cell neoplasms, further confirming it as the current gold standard [18]. In addition, the lack of immunolabeling for SOX-10 can be used to exclude melanocytic origin, as the antibody had 100% sensitivity for detecting melanocytic neoplasms in that same study [18]. However, positive labeling for SOX-10 does not distinguish between soft tissue sarcomas and melanocytic neoplasms, as the antibody labels a percentage of canine soft tissue sarcomas [18]. Lastly, evaluation for RNA levels of TYR, CD34, and CALD may help to further differentiate between canine oral amelanotic spindloid malignant melanomas and oral soft tissue spindle cell sarcomas [18].

**Figures 1–7 vetsci-09-00175-f001:**
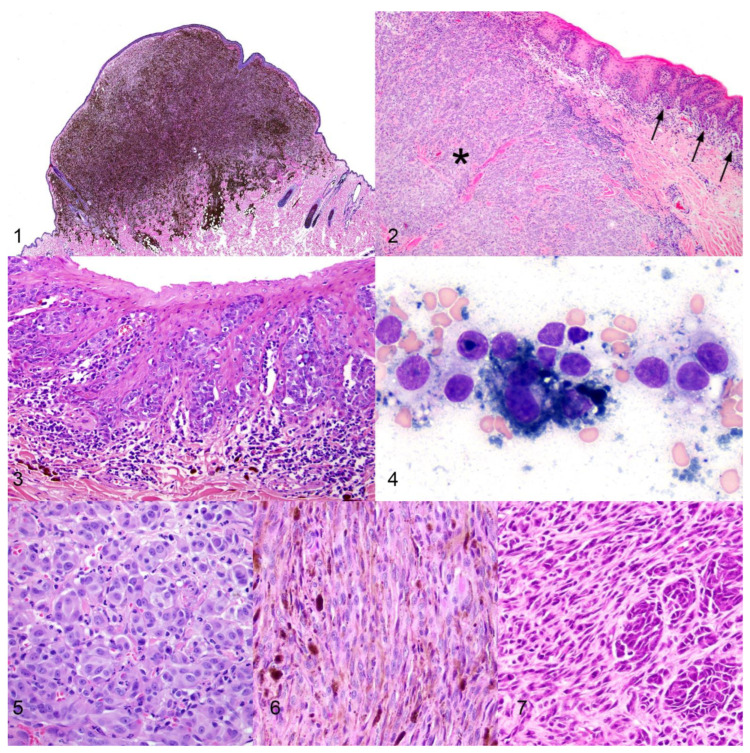
Histologic and cytologic features of canine melanocytic neoplasms. **Figure 1.** A raised, well circumscribed cutaneous melanocytoma that is localized in the superficial dermis. H&E. **Figure 2.** Malignant oral melanoma (compound melanoma) with vertical growth (*) and lentiginous spread (arrows). H&E. **Figure 3.** Junctional activity in an oral malignant melanoma. H&E. **Figure 4.** Fine-needle aspirate of round to polygonal neoplastic melanocytes that contain dark blue to black melanin pigment. Wright’s Giemsa. **Figures 5–7.** Cell morphology can be highly variable in canine melanocytic neoplasms, with epithelioid (Figure 5), spindloid (Figure 6), and mixed epithelioid and spindloid (Figure 7) subtypes being most common. H&E.

**Figures 8–13 vetsci-09-00175-f002:**
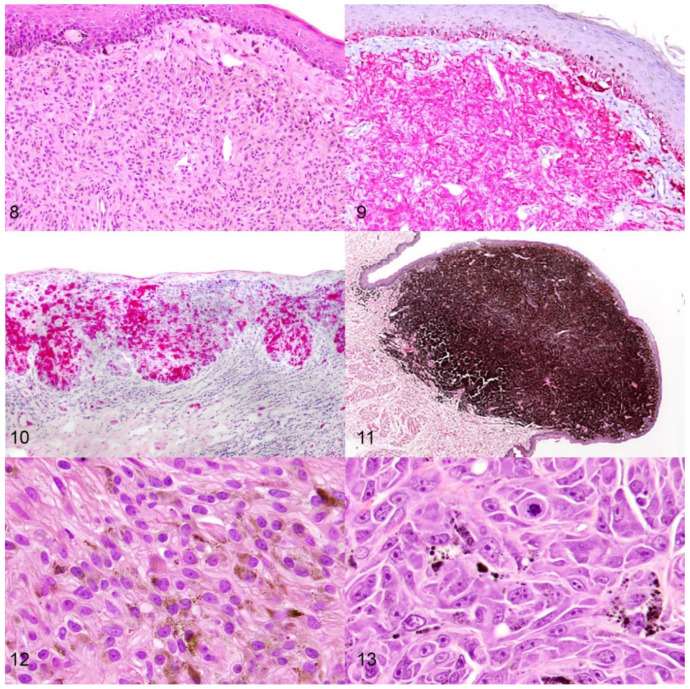
Diagnostic and prognostic features of canine melanocytic neoplasms. **Figures 8 and 9.** Oral spindle cell neoplasms that are present at the epithelial–subepithelial junction but lack clear pigmentation or junctional activity (Figure 8, H&E) require immunohistochemistry with an antibody cocktail against Melan-A, PNL-2, TRP-1, and TRP-2 for an accurate diagnosis (Figure 9, red chromogen, hematoxylin counterstain). **Figure 10.** Intraepithelial neoplastic melanocytes are commonly strongly positive with an antibody cocktail against Melan-A, PNL-2, TRP-1, and TRP-2. Red chromogen, hematoxylin counterstain. IHC. **Figure 11.** Oral melanocytic neoplasms of low malignant potential present as heavily pigmented, non-ulcerated, raised oral masses. H&E **Figures 12 and 13.** Nuclear atypia is an important prognostic criterion with well-differentiated neoplastic cells having small nuclei and a single centrally oriented nucleolus (Figure 12) and poorly differentiated neoplastic cells having a higher degree of anisokaryosis with larger, often multiple, nucleoli (Figure 13). H&E.

**Figure vetsci-09-00175-f003:**
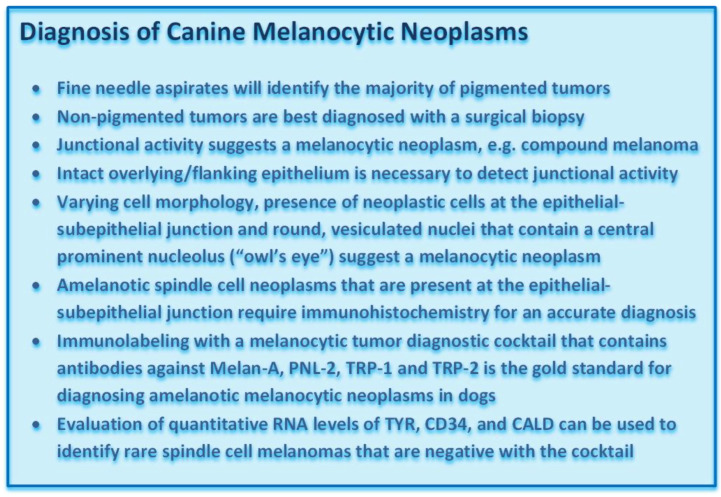


## 3. Prognosis of Canine Melanocytic Neoplasms

Several different parameters are used to prognosticate canine melanocytic neoplasms. Few of these parameters are only prognostically relevant for either cutaneous or oral melanocytic neoplasms. Histologic and molecular parameters are the most useful prognostic factors. Ultimately the Ki-67 index is thought to be the most reliable prognostic factor for canine melanocytic neoplasms [9,23].

### 3.1. Signalment

While signalment has been associated with the prognosis of canine melanocytic neoplasms, it should never be used as a standalone prognostic factor, as specificity is low. Some breeds are more likely to develop benign melanocytic tumors, and others are more likely to develop malignant forms. One study reported that in Doberman Pinschers and Miniature Schnauzers 75% of melanocytic neoplasms exhibited benign behavior, and in Miniature Poodles, more than 85% of melanocytic neoplasms exhibited malignant behavior [24]. Older dogs may be more prone to developing malignant melanomas than younger dogs [24]. One study demonstrated age negatively influenced survival of dogs with melanocytic neoplasms of the lips, feet, and skin [25]. However, due to a higher likelihood of perioperative complications and comorbidities in older patients, the independence of age as a prognostic parameter is difficult to evaluate [9]. Sex has not been associated with outcome [9].

### 3.2. Location

In general, melanocytic neoplasms of the nail bed, at mucocutaneous junctions, or in the oral cavity have a worse prognosis than cutaneous melanocytic neoplasms located elsewhere [9,25,26]. One study reported that cutaneous melanocytic neoplasms had the longest median survival time, while melanocytic neoplasms of the nail bed and mucocutaneous junctions had intermediate survival times, and oral melanocytic neoplasms had the shortest survival times [25]. Another study also showed that nail bed melanocytic neoplasms tended to be malignant, and cutaneous melanocytic neoplasms of the abdomen were more often benign [26]. However, cutaneous malignant melanomas can occur at all sites, and less aggressive melanocytic neoplasms may arise in the nail bed, mucocutaneous junctions, or oral cavity. For example, a histologically distinct subset of oral and lip melanocytic neoplasms that exhibits less aggressive biological behavior, called well-differentiated oral/lip melanocytic neoplasms or melanocytic neoplasms of low malignant potential, has been identified [10,12,23,27]. Therefore, location also lacks specificity as a sole prognostic factor.

### 3.3. Histologic Parameters

Nuclear atypia, mitotic count (number of mitoses in a field area of 2.37 mm^2^) [28,29] degree of pigmentation, level of infiltration, and vascular invasion are currently the most useful histologic parameters to predict the prognosis of both cutaneous and oral/lip melanocytic neoplasms [5,6,23,25,27,30,31,32,33,34,35]. Ulceration and tumor thickness, as determined by measuring a histologic section of the mass on an HE stained slide, are also useful prognostic factors for cutaneous melanocytic neoplasms [5,9,30,34].

There are a number of histologic criteria that allow pathologists to distinguish canine oral melanocytic neoplasms of low malignant potential from aggressive oral malignant melanomas. Oral melanocytic neoplasms of low malignant potential most closely resemble blue nevi in people [10,12,27]. They present as heavily pigmented, non-ulcerated, raised, or pedunculated oral masses that typically measure less than 1.0 cm in diameter [Figure 11] [27]. The heavily pigmented neoplastic cells form symmetric, wedge-shaped masses in the mid and upper subepithelial stroma that are divided by variably dense fibrous bands. Neoplastic cells are uniform, round or elongated, and contain a small round nucleus that contains a small, single, centrally placed nucleolus [27]. Anisokaryosis and anisocytosis are mild, and the mitotic count is low. Junctional activity or lentiginous spread is uncommon. In contrast, canine oral malignant melanomas are generally compound melanomas with junctional activity and deep nodular vertical growth. They closely resemble human mucocutaneous melanomas of the rectum or nasopharynx [12,27]. The degree of pigmentation is highly variable and amelanotic neoplasms are common. Neoplastic cells often have highly variable morphology, ranging from spindloid to epithelioid to round cell morphology, and mixed morphologic patterns are common. Anisokaryosis and anisocytosis are moderate to marked, and the mitotic count is high.

Cutaneous melanocytomas are generally small, raised, non-ulcerated, heavily pigmented neoplasms that are often confined to the dermis. Junctional activity is less common in cutaneous melanocytic neoplasms than in oral melanocytic neoplasms but can be present. There are varying data on junctional activity and prognosis [9]. Neoplastic cells are well-differentiated, polygonal to spindloid, often contain abundant melanin pigment, and have round, uniform, finely stippled nuclei that contain one prominent nucleolus. Mitoses are rare. Cutaneous malignant melanomas are generally larger, ulcerated, extend beyond the dermis to the deeper tissues, and are poorly pigmented. Epidermal involvement may or may not be present. Neoplastic cells vary in cellular morphology, have more atypical nuclei, and more frequent mitoses.

#### 3.3.1. Nuclear Atypia

Nuclear atypia is determined according to the criteria established by Spangler and Kass for both cutaneous and oral/lip melanocytic neoplasms [Figures 12 and 13] [25]. While assessment of nuclear atypia is somewhat subjective, using these strict criteria reduces interobserver variation [25]. Nuclear atypia can be assessed in melanocytic neoplasms with a predominant epithelioid subtype and in the spindloid subtype if there is sufficiently observable nuclear detail [23,25]. Bleaching may be necessary in heavily pigmented melanocytic neoplasms to allow for better visualization of the nucleus. At 40× magnification, 100–200 cells are counted and scored as well-differentiated or less differentiated. A threshold of 20% atypical nuclei in cutaneous melanocytic neoplasms and a threshold of 30% atypical nuclei in oral/lip melanocytic neoplasms differentiates aggressive from less aggressive tumors [23,25,34,35].

#### 3.3.2. Mitotic Count

The mitotic count has been shown to be a reliable prognostic indicator for canine cutaneous and oral melanocytic neoplasms in numerous studies [5,6,9,26,27,30,31,32,33,34,36,37]. However, interobserver variation can be high, as it may be difficult to differentiate mitotic figures from karyorrhectic nuclei, and identification of the area of highest mitotic density within a neoplasm can be challenging. Melanocytic neoplasms should be scanned at a 10× magnification to identify the area of highest mitotic activity. Bleaching may be required in heavily pigmented neoplasms. This is especially helpful for identifying “hot spots” when scanning the tumor at lower magnifications. It is recommended to report if the mitotic count was determined with or without bleaching [29]. Areas of ulceration, necrosis, and inflammation should be avoided, and mitoses are counted in 10 consecutive hpfs. The mitotic count should be reported as the number of mitoses in a field area of 2.37 mm^2^, which equals 10 fields at 40× magnification with an FN 22 ocular [28,29]. Studies prior to 2016 reported mitotic count as the number of mitoses in 10 hpf and did not always specify the FN of the ocular. Thus, the field area may have varied slightly between studies. However, the most commonly used ocular by pathologists in general has an FN of 22. Thus, there is a good chance that most studies actually were evaluating an area of 2.37 mm^2^. Regardless, even if a slightly differently sized ocular was used in some of the previous studies, the thresholds would not be expected to be affected significantly. For cutaneous melanocytic neoplasms, a mitotic count of 3 or more has been associated with shorter survival times compared to neoplasms with a mitotic count less than 3 [5,16,30,34]. For oral and lip melanocytic neoplasms, a mitotic count of 4 or more has been associated with shorter survival times compared to neoplasms with a mitotic count of less than 4. This cutoff was shown to have a sensitivity of 90% and a specificity of 84% in one study [23].

#### 3.3.3. Pigmentation

Pigmentation is somewhat difficult to quantitate histologically, resulting in interobserver variation. Regardless, highly pigmented cutaneous and oral/lip melanocytic neoplasms do appear to have longer survival times, and a cutoff of equal to or more than 50% of cells being pigmented has been associated with a better prognosis [9,23,30,33,34]. In contrast, no or low pigmentation has not been significantly associated with a poor prognosis [9,23,30]. Therefore, the degree of pigmentation should not be used as a standalone prognostic factor.

#### 3.3.4. Level of Infiltration

The level of infiltration of neoplastic melanocytes into the underlying stroma has been shown to be prognostically significant in both cutaneous and oral/lip melanocytic neoplasms [8,30,37,38]. It is recommended that the deepest level of tissue infiltration in cutaneous melanocytic neoplasms be reported as: epidermis only, superficial dermis (superficial dermal vascular plexus), middle dermis (mid dermis vascular plexus), deep dermis (deep dermal vascular plexus), subcutis, skeletal muscle (cutaneous trunci or regional muscle), or bone [29]. The cutaneous melanocytic neoplasms that extend beyond the dermis tend to exhibit more aggressive biological behavior [30]. For oral melanocytic neoplasms, it is recommended that the deepest level of tissue infiltration be reported as: mucosal epithelium only, submucosa, skeletal muscle, salivary gland, or bone [29]. Those oral melanocytic neoplasms that are superficial and raised and do not involve bone tend to show less aggressive behavior than those that extend deeper or invade bone [8,37,38]. Accurate assessment of this parameter requires submission of clean deep margins, and the lack thereof should be reported by the pathologist. Regardless, the level of infiltration should also not be used as a standalone parameter.

#### 3.3.5. Vascular Invasion

Blood or lymphatic vascular invasion is a strong prognostic indicator of malignant behavior for both cutaneous and oral/lip melanocytic neoplasms [11].

#### 3.3.6. Ulceration

The presence of ulceration is used to predict more aggressive behavior of cutaneous melanocytic neoplasms only [30]. It is not a standalone parameter, as it can be easily impacted by a variety of external factors, such as self-trauma [9].

#### 3.3.7. Tumor Thickness

Tumor thickness is a more recently recognized prognostic parameter for cutaneous melanocytic neoplasms [5,34]. Tumor thickness is measured in HE stained slides along a perpendicular line from the epidermis to the deep margin of the neoplasm through the area of its greatest thickness [34]. The thickness can be measured with either an ocular micrometer or a ruler. A tumor thickness of more than 0.95 cm has been associated with a shorter overall survival time and a higher risk of death, and a tumor thickness of more than 0.75 cm has been associated with a shorter disease-free time and a higher risk of recurrence or metastasis [34].

**Figure vetsci-09-00175-f004:**
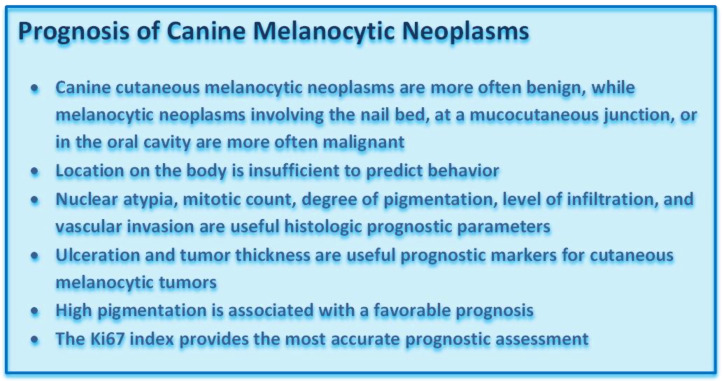


### 3.4. Molecular Parameters

Several molecular parameters have been evaluated as prognostic markers but only the Ki-67 index is currently extensively used in a diagnostic setting.

#### Ki-67 Index

Several studies have demonstrated that the Ki-67 index, as determined by IHC, predicts the prognosis of both cutaneous and oral/lip melanocytic neoplasms [5,9,23,30,36,37,39,40]. More importantly, this parameter has the highest degree of reproducibility compared to the previously described markers and is least affected by interobserver variability. Ki-67 reflects the growth fraction of a neoplastic cell population and detects cells in all phases of the cell cycle, except the resting phase. For canine cutaneous and oral/lip melanocytic neoplasms, the Ki-67 index is assessed in non-ulcerated and non-inflamed regions of the neoplasm [Figures 14 and 15]. Bleaching may be necessary after IHC labeling is performed to better visualize nuclei [9]. The number of Ki-67 labeled nuclei is determined in the area of highest labeling.

For cutaneous melanocytic neoplasms, the number of positive nuclei in 500 cells is counted at a magnification of 40× and reported as a percentage of positive cells [5,30]. A 1 cm^2^ optical grid reticle can be placed into the eyepiece and is helpful to keep track of which cells have been counted. At 40× magnification, 1 cm/40 = 0.25 mm (0.0625 mm^2^). Five fields = 0.3125 mm^2^, and 10 fields = 0.625 mm^2^. A threshold of 15% or more Ki-67 positive cells has been associated with shorter survival times of dogs with cutaneous melanocytic neoplasms [5,30]. None of the benign cutaneous melanocytic neoplasms had a Ki-67 index of 15% or more in one study [30]. The Ki-67 index has been shown to predict the prognosis of canine cutaneous melanocytic neoplasms more accurately than the mitotic count or other histologic criteria [30].

For oral/lip melanocytic neoplasms, the Ki-67 index is reported as the average number of positively labeled neoplastic melanocyte nuclei per area of a 1 cm ^2^ optical grid reticle at a magnification of 40× (5 grid areas counted) in the highest labeling area [23]. A threshold of 19.5 positive nuclei per grid area or more has been associated with lower survival times [23]. The Ki-67 index has been shown to more accurately predict the prognosis of canine oral/lip melanocytic neoplasms than the mitotic count or nuclear atypia [23].

**Figure vetsci-09-00175-f005:**
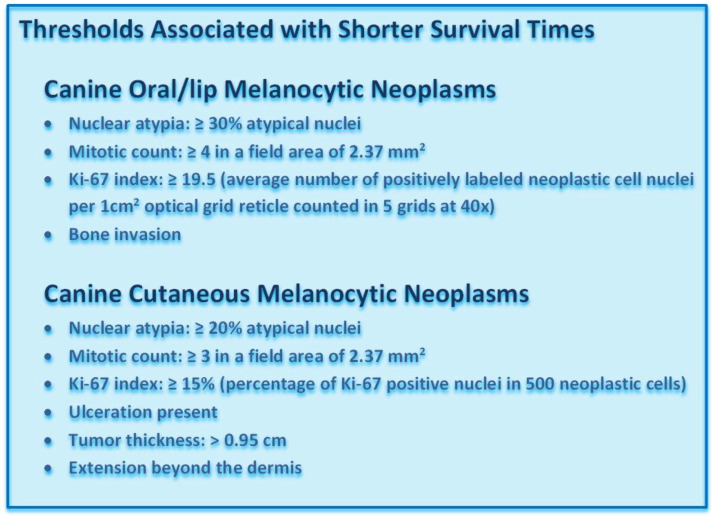


### 3.5. Margin Evaluation

While the above-described prognostic factors help to predict the overall biological behavior of melanocytic neoplasms, accurate assessment of surgical margins is an essential component for determining the likelihood of local recurrence [29,41]. Excisional biopsies should include wide lateral margins that include the lateral flanking epithelium, in order to increase the likelihood of an accurate diagnosis and of complete excision. Wide excision of the lateral flanking epithelium is especially important in oral compound melanomas [10,12]. Submission of the intact flanking epithelium does not only increase the diagnostic accuracy, as the pathologist can examine these sections for junctional activity, but it also increases the likelihood of complete excision, as lentiginous spread is a common feature of oral malignant melanomas that is generally not observed on gross examination [12,18].

While routine microscopic examination is most commonly based on cross-sectioning (half and radial sections) of the submitted neoplastic mass, such sectioning is insufficient to detect laterally escaping neoplastic cells, especially in the overlying epithelium [41]. Accurate margin assessment requires a combination of the cross-sectioning method with tangential sectioning of tumor margins to confirm complete surgical excision of canine melanocytic neoplasms [Figure 16] [41]. For oral melanocytic neoplasms, the underlying bone should be evaluated for tumor invasion via imaging and histopathology when appropriate [Figure 17] [41]. Similarly, for nail bed melanocytic neoplasms, the digit should be amputated, and proximal soft tissue margins should be examined [41]. After decalcification, the proximal bone margins, as well as longitudinal bone sections, should be examined [Figure 18] [41].

**Figures 14 and 15 vetsci-09-00175-f006:**
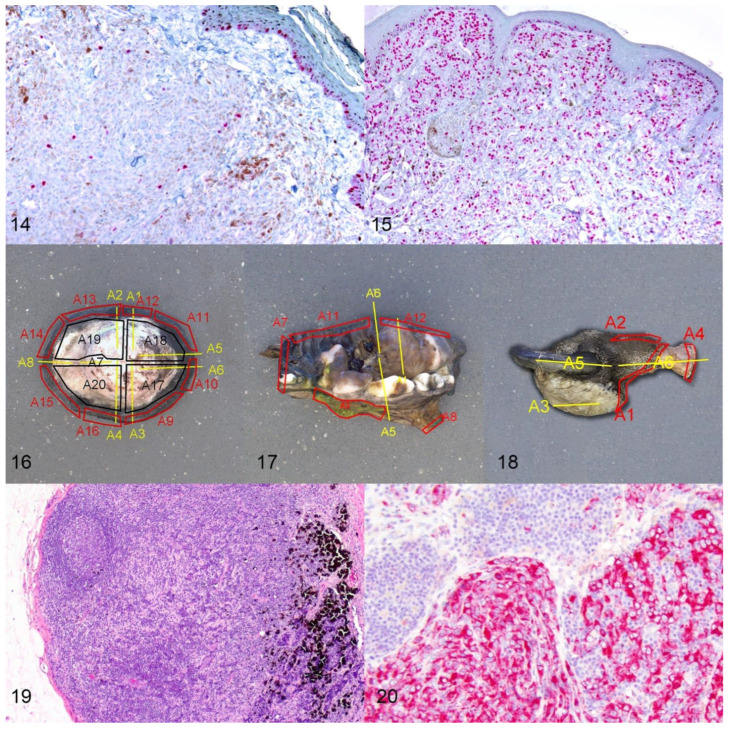
The Ki-67 index is the most accurate predictor of prognosis. Melanocytic neoplasms with a low Ki-67 index (Figure 19) have significantly longer survival times than those with a high Ki-67 index (Figure 15). Nuclei are labeled with a red chromogen, hematoxylin counterstain. **Figures 16–18.** Margin evaluation of canine melanocytic neoplasms. A combination of cross-sectioning (radial sections, A1–A8, yellow) and tangential sectioning (lateral tangential, A9–A16, red; deep Table A17. A20, black) is best to evaluate margins in cutaneous melanocytic neoplasms (Figure 16). For oral melanocytic neoplasms, cross-sections of the mass (A1 yellow), tangential sections of the soft tissue margins (A11–A12, red), and after decalcification cross-sectioning of the underlying bone (A5–A6 yellow) and sectioning of bone margins (A2, A7–A8 red, others not shown) are recommended (Figure 17). For tumors of the nail bed, cross-sections through the tumor (A3, yellow) and proximal soft tissue (A2–A3, red) and bone margins (A4, red) and longitudinal bone sections (A5–A6, yellow) should be examined (Figure 18). **Figures 19 and 20.** Assessment of nodal spread. Differentiating between melanophages, draining melanocytes, or neoplastic melanocytes can be challenging, especially in lymph nodes (Figure 19), and immunohistochemistry for Melan-A or PNL-2 (Figure 20) can at least distinguish macrophages from melanocytes.

**Figure vetsci-09-00175-f007:**
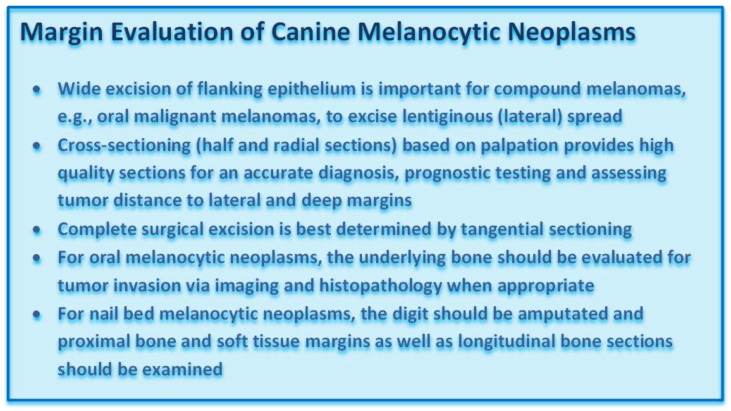


### 3.6. Lymph Node Assessment

Evaluation of sentinel lymph nodes for metastasis of melanocytic neoplasms is an important component of staging and determination of treatment strategies. While the mandibular lymph node is the primary metastatic site for canine oral/lip melanocytic neoplasms, 50% of these tumors commonly metastasize to other lymph nodes, especially retropharyngeal lymph nodes [42]. Contralateral dissemination has been shown to occur in 62% of affected dogs while ipsilateral metastasis occurs in 92% [42,43,44,45]. Determining the sentinel node through radioactive dye during surgery will increase the likelihood of detecting and removing nodal metastases; however, there is inconsistency among veterinarians in performing elective neck dissection [46]. Computed tomography (CT) lymphangiography has been found insufficient to detect sentinel nodes, while use of 18 fluorine-fluorodeoxyglucose (18 F-FDG) positron emission tomography/computed tomography (PET/CT) with a standardized uptake value (SUV) max cut point of 3.3 had 100% sensitivity and 83% specificity of detecting nodal metastasis [47,48].

The detection of metastatic melanoma by cytology, or histology, can be challenging, but there are some methods that pathologists can use to improve the sensitivity and specificity of metastasis detection. Fine-needle aspiration and cytology of lymph nodes is an easy, minimally invasive way to detect metastasis. However, the sensitivity of this method is low if there is only micrometastasis, as it can be easy to miss the areas of neoplastic cells during aspiration. The consensus between routine cytology and histopathology for staging lymph nodes in patients with melanocytic neoplasms is poor [49]; however, in one study, there appeared to be increased detection of amelanocytic forms with ICC for Melan-A on fine-needle aspirate samples, particularly when cells were low in number or of round cell morphology [15,50]. Histologic evaluation is the preferred method, but unless there is overt metastasis, it still can be easy to miss an area of micrometastasis during routine trimming. The first problem is cost. Veterinary laboratories generally only evaluate 1–3 sections of a submitted lymph node, as opposed to complete bread-loafing, which is performed in human medicine. Bread-loafing or step-sectioning at 0.2 cm intervals has been recommended for nodal evaluation in a number of human and canine neoplasms [51,52,53]. Thus, it is very easy to simply miss micrometastases in the examined sections. Second, it can be difficult to differentiate between hemosiderin and melanin in some cases [Figure 19]. An iron stain, such as Prussian blue, can assist with this. The third difficulty is determining if the pigmented cells are melanophages, draining melanocytes, or neoplastic melanocytes. Even normal lymph nodes may contain melanocytes, melanophages, or pigment granules, especially mandibular nodes from dogs with highly pigmented oral epithelium [11,42,54,55]. The best way to distinguish between melanocytes and macrophages is to perform IHC for a melanocyte marker such as Melan-A or PNL-2 or to perform the melanocytic diagnostic cocktail [Figure 20]. Aberrantly labeled plasma cells need to be discounted. Lack of labeling does not completely rule out melanocytic origin, but if they are heavily pigmented, they are likely well-differentiated and should have a very high likelihood of expressing one of the described melanocyte markers. Just like with mast cells, it is not always possible to differentiate between neoplastic and non-neoplastic melanocytes, but features such as disruption of the normal lymph node architecture and cellular and nuclear atypia indicate neoplasia.

**Figure vetsci-09-00175-f008:**
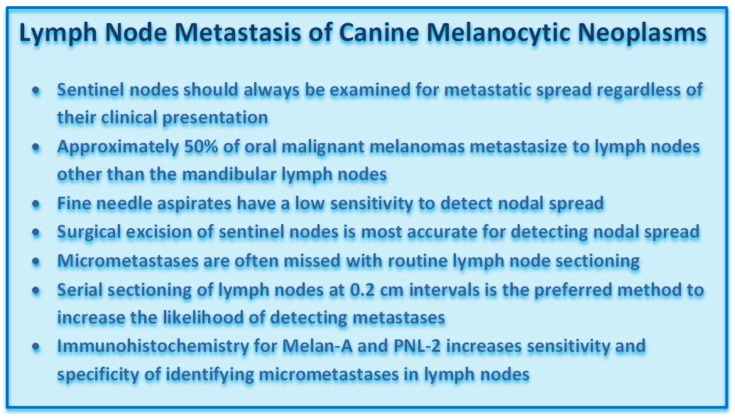


## 4. Conclusions

Canine melanocytic neoplasms have variable biological behavior regardless of their location and prognostic evaluation requires a combination of microscopic assessment for nuclear atypia, mitotic count, degree of pigmentation, level of infiltration, and vascular invasion along with immunohistochemical determination of the Ki67 index. For cutaneous melanocytic neoplasms, the tumor thickness and ulceration of the lesion should be included in such evaluation. The general diagnosis of canine melanocytic neoplasms is often accomplished with fine-needle aspirates, but histopathology is often needed to differentiate between benign and malignant forms. Surgical biopsy is most important for oral melanocytic neoplasms as they are often compound melanomas with wide lateral intraepithelial (lentiginous) spread. Surgical excision should account for this characteristic feature, and submission of intact epithelium flanking an ulcerated area over the primary mass is strongly recommended. Wide excision also increases diagnostic accuracy by allowing the pathologist to search for junctional activity. Furthermore, amelanotic spindloid melanomas have to be differentiated from soft tissue sarcomas. An immunohistochemical cocktail that contains antibodies against Melan-A, PNL-2, TRP-1, and TRP-2 is the current gold standard to identify melanocytic neoplasms. RNA expression analysis for TYR, CD34, and CALD can be used in some challenging cases to confirm the diagnosis. Radial and tangential sectioning should be combined to provide the most accurate margin assessment. As oral malignant melanomas readily metastasize to the tributary lymph nodes, sentinel lymph nodes should be evaluated when a malignant oral melanoma is suspected. Microscopic evaluation of such nodes is best performed by serial sectioning of the node combined with immunohistochemical labeling for Melan-A and PNL-2.

## Data Availability

Not applicable.

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
