# Peer review of "Diagnosis and Prognosis of Canine Melanocytic Neoplasms"

_vetsci, 2022, doi:10.3390/vetsci9040175_

Round 1

Reviewer 1 Report

This review is overall informative and contains a complete scan of the available literature on diagnostic and prognostic knowledge on canine melanocytic tumors, which is useful for both research and diagnostic environment. The review is well organized and all the information is correctly referenced. 

There are minor issues should be fixed before publication, in my opinion.

Figure 12 legend: check "anysokaryois" spelling

Page 8 line 207: we currently assess the mitotic count as defined by Meuten and colleagues, but some of the studies were performed some years prior to this definition. Therefore, I don't know if it is correct to cite Meuten's et al. articles as the protocol at the beginning of the paragraph (even though I obviously agree with the use of a standardized method). In some of the papers reported at the end of the paragraph it is clearly stated that the mitotic count has been assessed as defined by Meuten, but what about the method adopted by studies prior to 2016 to assess mitoses? I guess that the value of mitoses in prognosis does not change that much and is still valid, but the methods to evaluate could be a bit different (ocular FN could have been smaller than 22 and no adjustment could have been considered). This is something I was wondering myself quite a lot of times when using the 3/4 mitoses threshold on 10 HPF.

In paragraph 3.5 again the "old" definition of mitotic index is mixed with mitotic count, probably a comment on this point could be helpful to clarify the differences.

In Ki-67 index paragraph, the use of a 1 cm2 optical grid is reported, but the details of the grid are missing. I think that this should be an important point to clarify, to help pathologists to standardize also the evaluation of ki-67 mitotic index.

Figure 19 legend: I am not aware that the term "melanomacrophages" can be applied also to the cells that I call "melanophages". I know that melanomacrophages are aggregates of macrophages containing melanin that can be found in some fishes' and reptiles' organs. If the authors know a different use of these definitions, I am happy to learn.

please correct " frommelanocytes"

page 13 line 397: see my comment on melanomacrophages

Author Response

This review is overall informative and contains a complete scan of the available literature on diagnostic and prognostic knowledge on canine melanocytic tumors, which is useful for both research and diagnostic environment. The review is well organized and all the information is correctly referenced.

Thank you for your thorough edits and helpful suggestions. We have responded to each request below. We would like to indicate that this paper broadly targets veterinary clinicians and diagnosticians, rather than only pathologists.

There are minor issues should be fixed before publication, in my opinion.

Figure 12 legend: check "anysokaryois" spelling

This has been corrected.

Page 8 line 207: we currently assess the mitotic count as defined by Meuten and colleagues, but some of the studies were performed some years prior to this definition. Therefore, I don't know if it is correct to cite Meuten's et al. articles as the protocol at the beginning of the paragraph (even though I obviously agree with the use of a standardized method). In some of the papers reported at the end of the paragraph it is clearly stated that the mitotic count has been assessed as defined by Meuten, but what about the method adopted by studies prior to 2016 to assess mitoses? I guess that the value of mitoses in prognosis does not change that much and is still valid, but the methods to evaluate could be a bit different (ocular FN could have been smaller than 22 and no adjustment could have been considered). This is something I was wondering myself quite a lot of times when using the 3/4 mitoses threshold on 10 HPF.

We hope that the following additions help to clarify mitotic count.

In this sentence: “The mitotic count is reported as the number of mitoses in a field area of 2.37 mm2, which equals 10 fields at 400x magnification with an FN 22 ocular.[28][29]” We are only stating how mitotic count should be reported, not how it has been reported in some older studies. We have changed the sentence to read: “The mitotic count should be reported as the number of mitoses in a field area of 2.37 mm2, which equals 10 fields at 400x magnification with an FN 22 ocular.[28][29]”

And we have added the following after this sentence:

“Studies prior to 2016 reported mitotic count as the number of mitoses in 10 hpf and did not always specify the FN of the ocular. Thus, the field area may have varied slightly between studies. However, the most commonly used ocular by pathologists in general has an FN of 22. Thus, there is a good chance that most studies actually were evaluating an area of 2.37 mm2. Regardless, even if a slightly different sized ocular was used in some of the previous studies, the thresholds would not be expected to be affected significantly.”

If a FN of 20 would have been used in an older study, the difference in the examined area would be only 17%. With cut-off values of 3 or 4 mitoses in this area, a difference in 20% area size examined is not going to change the outcome for an individual case. We believe that the larger problem of the mitotic count is the ability of the pathologist to detect the areas of highest mitotic density, especially in larger samples or severely inflamed samples. While pigmentation interference can be overcome by bleaching, recent studies have demonstrated with other tumors the high interobserver variation in finding the areas of highest mitotic density (Bertram et al. 2022, https://pubmed.ncbi.nlm.nih.gov/34965805/). This problem is even more prevalent in scanned slides that don’t allow focusing up and down to differentiate mitoses from other structures. There have been numerous studies that show an underestimation of mitotic counts in scanned slides (for review Donovan et al. 2022, https://pubmed.ncbi.nlm.nih.gov/33371818/) compared to light microscopy. Not only is Ki67 more accurate for assessing proliferation activity as it determines the growth fraction of a neoplastic cell population rather than only the phase index as the mitotic count does, it is also much easier to quickly identify positive nuclei when using a red chromogen on a blue (hematoxylin) background.  

In paragraph 3.5 again the "old" definition of mitotic index is mixed with mitotic count, probably a comment on this point could be helpful to clarify the differences.

We are sorry but we are not sure what you are referring to here. Once we defined mitotic count as number of mitoses in a field area of 2.37 mm2 we simply stick with that throughout the paper. We did add a line above that discusses the misuse of the term of “mitotic index”.

In Ki-67 index paragraph, the use of a 1 cm2 optical grid is reported, but the details of the grid are missing. I think that this should be an important point to clarify, to help pathologists to standardize also the evaluation of ki-67 mitotic index.

We assume that you mean Ki-67 index and not Ki-67 mitotic index. Our target audience for this review is veterinary clinicians so we did not aim to provide to much technical detail in some areas. We did add: “A 1 cm2 optical grid reticle can be placed into the eyepiece and is helpful to keep track of which cells have been counted. At 40x magnification, 1 cm/40 = 0.25 mm (0.0625 mm2). Five fields = 0.3125 mm2 and 10 fields = 0.625 mm2.” We can cite this webpage if desired that provides the most recent prognostic protocol for canine oral melanocytic neoplasms: https://cloud.cldavis.org/index.php/s/LkWtFmZdE8KxxQy  This webpage also contains a link to the actual grid information: https://bolioptics.com/microscope-eyepiece-reticle-net-grid-micrometer-10x10mm-100-squares-dia-22mm-1-5mm/ .

Figure 19 legend: I am not aware that the term "melanomacrophages" can be applied also to the cells that I call "melanophages". I know that melanomacrophages are aggregates of macrophages containing melanin that can be found in some fishes' and reptiles' organs. If the authors know a different use of these definitions, I am happy to learn.

We agree that melanophages is the more correct term and we have made this change.

please correct " frommelanocytes"

This has been corrected.

page 13 line 397: see my comment on melanomacrophages

See above

Reviewer 2 Report

This is a well-organized review of the most validated features for the diagnosis and prognosis of canine melanocytic tumours, which was sought in the field. Sometimes the authors went deeper in the description of oral melanocytic tumours compared to the cutaneous ones, but anyway, the information provided is clear, correct and useful. Although I suggested several changes/clarifications, I cannot see any major flaws. Hence, I recommend publication after minor revision.

Author Response

OVERVIEW AND SUMMARY
In the review, the authors aimed to focus on the current validated factors useful for the diagnosis and the prognosis of canine melanocytic neoplasms. This is an important topic, as from the last review on the same topic (in 2011) several research papers were published and after the last consensus paper (August 2020) a review with an update of the current knowledge was desirable. The review is well organized and the relevant literature is cited; figures are of high quality and appropriate. However, there are some minor points to be edited/clarified to improve the general clarity and readability of the manuscript.

Thank you for your thorough edits and helpful suggestions. We have responded to each request below. We would like to indicate that this paper broadly targets veterinary clinicians and diagnosticians, rather than only pathologists.

MINOR COMMENTS
Introduction:
line 31-35:Talking about prognostication, you wrote only about “dermal melanocytic neoplasms”. Although they are the most common cutaneous melanocytic tumours in the dog, they can also be compound or junctional (Tumors in domestic animals, Meuten 2017; Jubb, Kennedy and Palmer’s Pathology of Domestic Animals, Maxie 2016). Why do not talk about cutaneous melanocytic tumours in general? Could you please clarify this point?

Thank you for catching this wording. We actually used the term “dermal” to simply mean “cutaneous” in a few areas, but you are right, this is not correct. We have changed the term dermal to cutaneous throughout the text except where we are defining the terms “dermal” and “compound”.

lines 41-45: this sentence interrupts the flow of this part of the paper. You started talking about
terminology and continued at line 46. Please, move it (I would suggest after morphological features).

We opted to reword lines 41-45 and just state: For routine surgical pathology, the term“melanocytic neoplasm” should be used when the histologic parameters alone cannot differentiate between a melanocytoma and a malignant melanoma.[7][9]
This hopefully addresses your concern.

lines 52-53: Did you mean “cutaneous melanocytic tumours” instead of “dermal”, at the beginning of the sentence?

As explained earlier, we made changes throughout the manuscript and the terms dermal and cutaneous throughout the manuscript should now be accurately applied.

lines 53-62: in reference 9 there is no mention of the vertical and radial growth, nor lentiginous and
pagetoid spread, so please remove the reference when in correlation with those terms.

It has been removed.

Diagnosis of canine melanocytic neoplasms:
line 83: “(e.g. Wright’s Giemsa)” should be moved after “stains”, since it interrupts the sentence.

This has been removed.

lines 87-88: the sentence would be clearer if specify: “...to differentiate between these non-melanocytic
tumours or to diagnose amelanotic melanocytic neoplasms”

This suggestion has been adopted.

lines 89-93: till this point, you were talking about pigmented tumours, hence introducing here the
amelanotic ones is confusing. Additionally, since in this part you are describing cytology, jumping to
histopathology and coming back to cytology create confusion. It is better to move this sentence as an
introduction to the part about histology. Moreover, the last lines (91-93) are unclear: what do you mean by “cytologic examination of histologically amelanotic neoplasms”? Did you mean the morphology of cells seen in histological preparations?

We attempted to clarify this. We are trying to describe the different ways one can go about to reach a diagnosis of melanocytic origin for amelanotic tumors. We hope this addresses your concerns:

If Melan-A ICC is negative and amelanotic melanoma is still strongly suspected, histopathology is recommended, as it generally allows for more accurate diagnosis, via assessment of other morphologic features and the ability to perform additional immunohistochemical markers, and also provides an accurate prognosis.[15] Alternatively, for some histologically amelanotic melanocytic neoplasms that do not exhibit other confirmatory features of melanocytic origin, cytology may be the more sensitive method for detecting melanin pigment. However, in some cases, melanin may not be seen cytologically either.

lines 119-122: the epithelioid, spindloid or mixed morphology, as stated at the beginning of the sentence, are anyway the most common predominant cell type also in less differentiated neoplasms. Therefore, they are not “more likely” to resemble other cell types, simply other cellular morphologies can be seen in these types of tumours. Please modulate this sentence.

We have changed this to:

While epithelioid/polygonal, spindloid/fibromatous, and mixed epithelioid and spindloid are still the most common cellular morphologies of poorly differentiated melanocytic neoplasms, other cellular morphologies can also be seen in these  less differentiated tumors.[7][9]

line 135: the sentence is not concluded: “[...] IHC for the melanocytic specific markers has to be
performed”, perhaps?

Perhaps the colon was confusing? It now reads:

If a diagnosis cannot be reached on routine microscopic examination, immunohistochemistry (IHC) for the melanocyte specific markers Melan-A, PNL-2, tyrosine reactive protein (TRP)-1 and TRP-2 is most commonly used to confirm melanocytic differentiation [Figure 9].[3][18-22]

lines 141-143: do you mean that while the cells in the tumour mass can be negative, the intraepithelial
ones are positive? Could you please explain it better?

The sentence now reads:

While the subepithelial neoplastic cells may be negative, intraepithelial nests of neoplastic cells are the most commonly positive cells detected by IHC, further emphasizing the importance of submitting specimens with intact epithelium [Figure 10].[12][18]

lines 157-159: this study about RNA quantification is a quite new one. Could you please expand this part
explaining better how RNA levels differ in oral amelanotic melanomas and soft tissue sarcomas?

As we intended this manuscript for veterinary clinicians we elected not to go into as much technical detail in the molecular areas. Interested readers can refer to the Tsoi et al reference.

Prognosis of canine melanocytic neoplasms:
Histological parameters: in this part of the review, it would be more appropriate to list the different sub-
paragraphs (nuclear atypia, mitotic count, and so on) as 3.3.1, 3.3.2 etc..., instead of the consecutive
numeration given, since they are all histological parameters. The numeration should go on with 3.4 when arrived at the molecular parameters (in the same way the “Ki67 index” should be the number 3.4.1).

This change has been made.

lines 213-231: this part is focused on oral tumours and summarizes some morphological aspects of tumours with low and high malignant potential, including the gross morphology. It is useful and provides additional information and I would strongly suggest adding a similar paragraph for cutaneous tumours. I suggest removing the sentences about the mitotic count, because you will talk deeply about it later and it would be redundant (or provide also all the other key thresholds for prognosis). Probably, it would be better to start this section about histological parameters with these summarized descriptions.

We deleted the mitotic count thresholds here and just put low or high as more details are provided in subsequent paragraphs as you indicated.

We also added a section on cutaneous tumors:

While the subepithelial neoplastic cells may be negative, intraepithelial nests of neoplastic cells are the most commonly positive cells detected by IHC, further emphasizing the importance of submitting specimens with intact epithelium [Figure 10].[12][18]

lines 241-243: the thresholds are inverted: for cutaneous melanocytic neoplasms is 20%, while for oral/lip tumours is 30% (Smedley et al 2011, Bergin et al 2011). The same mistake is present in the box “Thresholds associated with shorter survival times” (page 11)

Thank you for catching this error!

line 256: “400x magnification” is a different way to indicate the hpf compared to the rest of the paper.
Please, to avoid confusion, consistently use the same method for indicating the magnification through the paper.

We used 40x throughout and specified the area as 2.37 mm2 were indicated

line 261: the reference is the same in the next sentence. It would interrupt less the reading if it was kept
only after this one.

This change has been made.

line 293: it would be useful to cite here Smedley et al 2011 (ref. 9), where a wider overview on this
parameter is furnished both in cutaneous and oral tumours.

This reference has been added.

line 295-298: for tumour thickness Laprie et al 2001 (ref 30) are cited. However, in this study there is no
mention of tumour thickness. Indeed the first paper introducing it is Lacroux et al 2012 (ref 5), although
they showed only the association with the clinical outcome and did not perform survival analysis. Please,
edit the reference. Additionally, since it is a quite new parameter, might be useful to insert that the
thickness can be measured with both an ocular micrometre and a ruler.

Thank you for catching this typo. It was meant to be the Lacroux reference. We also added the suggestion.

line 299: the 0.95 cm cut-off is associated with shorter overall survival and a higher risk of death (not
disease-free time). Please, correct it.

This has been corrected.

line 379: the number 18 is in superscript.

This has been corrected.

lines 400-402: considering the possibility of false-negative, why in lymph node evaluation do you suggest
using Melan-A or PNL-2 and not the cocktail as for the diagnosis?

It is simply cheaper to run the individual antibodies and melan-A and PNL-2 are the most sensitive. Most labs do not have cocktail but do have the individual antibodies. The cocktail could be performed as well. We added: or to perform the melanocytic diagnostic cocktail.

lines 408-410: it would be better to move this sentence as an introduction of the paragraph about cytology and histopathology, instead of being the conclusion. It seems to me the flow can be better if inserted after line 382.

This sentence has been moved.

Conclusions:
lines 414-419: as a logical step, the prognosis should be discussed after the diagnosis, which comes first.
Instead, you could start stating the challenge of diagnosing canine melanocytic tumours.
lines 420-421: “The diagnosis of canine melanocytic neoplasms is often accomplished with fine-needle
aspirates. However, surgical biopsy is most important especially in oral...”

We changed this to: The general diagnosis of canine melanocytic neoplasms is often accomplished with fine needle aspirates, but histopathology is often needed to differentiate between benign and malignant forms.

Figures:
line 68: there is a typo with the word “lentiginous”

This has been corrected.

line 166: “H&E” should be corrected with “IHC”

This has been corrected.

Figures 16-18: in the legend, please associate the colours for cross- and tangential sectioning. The alpha-
numeric code on the figures is unclear: please simplify it or clarify it in the legend.

We added colors and numbers to the margin explanations.

Boxes:
Diagnosis of Canine Melanocytic Neoplasms: (page 7)
- “oral amelanotic spindle cell neoplasms [...] require immunohistochemistry”. Why did you specify “oral”? All amelanotic melanocytic neoplasms require IHC for accurate diagnosis.

The change has been made

- “evaluation of quantitative RNA expression ...”: expression may let think about proteins. In this case might be better “RNA levels”, unless you prefer to use “gene expression”. The same at line 157.

The change has been made in the box and in the text.

Lymph node metastasis of canine melanocytic neoplasms: (page 14)
- “approximately 50% of oral malignant melanomas metastasize...”: this information is not provided in the text, please add it, with its reference.

This has been added.

- “serial sectioning of lymph nodes at 0.2 cm intervals...”: this information is not provided in the text, please add it, with its reference.

The information and references have been added.

Reviewer 3 Report

General comment: The authors presented an interesting and original work, concerning to the diagnosis and prognosis of canine melanocytic neoplasms.

The manuscript is written in a comprehensive way.

Title: The title short, concise, and adequate.

Abstract: It is adequate.

The keywords should be different from those used in the title.

Introduction: It is adequate. The authors provided an adequate overview of the thematic.

Figures: The quality of all figures should be improved. The magnification should be indicated for each figure. Some space among the figures will facilitate their observation and analysis.

Please add a title to the blue boxes and improve text quality (focus the letter).

Conclusion: The conclusion is adequate.

References: Please check the references citation in the text.

Recommendation: The manuscript should be accepted after a Moderate revision.

Author Response

Thank you for your thorough edits and helpful suggestions. We have responded to each request below.

General comment: The authors presented an interesting and original work, concerning to the diagnosis and prognosis of canine melanocytic neoplasms.

The manuscript is written in a comprehensive way.

Thank you

Title: The title short, concise, and adequate.

Thank you

Abstract: It is adequate.

The keywords should be different from those used in the title.

We are not aware of this particular rule and were under the impression that title and key words are independent, but are happy to follow the reviewers guidance. We would ask for the advice of the editor?

Introduction: It is adequate. The authors provided an adequate overview of the thematic.

Thank you

Figures: The quality of all figures should be improved. The magnification should be indicated for each figure. Some space among the figures will facilitate their observation and analysis.

Based on previous publishing experience with the journal, we are under the impression that the ultimate figure layout will be finalized by the editorial assistant. Most pathology journal have completely abandoned bars or magnification and this journal also leaves the choice with the authors. We believe that such bars will not add any information and rather distract from the image.

Please add a title to the blue boxes and improve text quality (focus the letter).

As each text box has a title on top adding an additional title/figure legend appears redundant. We will ask for guidance from the editorial assistant regarding the shading of the lettering and whether to change this.

Conclusion: The conclusion is adequate.

Thank you

References: Please check the references citation in the text.

Done

Recommendation: The manuscript should be accepted after a Moderate revision.

Thank you

Round 2

Reviewer 3 Report

The mansucript should be accepted for publication. 

Author Response

The mansucript should be accepted for publication. 

Thank you